# Nicotine Exposure in a Phencyclidine-Induced Mice Model of Schizophrenia: Sex-Selective Medial Prefrontal Cortex Protein Markers of the Combined Insults in Adolescent Mice

**DOI:** 10.3390/ijms241914634

**Published:** 2023-09-27

**Authors:** Andrés Rodríguez-Vega, Ana Carolina Dutra-Tavares, Thainá P. Souza, Keila A. Semeão, Claudio C. Filgueiras, Anderson Ribeiro-Carvalho, Alex C. Manhães, Yael Abreu-Villaça

**Affiliations:** 1Laboratório de Neurofisiologia, Departamento de Ciências Fisiológicas, Instituto de Biologia Roberto Alcantara Gomes, Universidade do Estado do Rio de Janeiro (UERJ), Rio de Janeiro 20550-170, RJ, Brazil; andresve22@gmail.com (A.R.-V.); dutratavaresana@gmail.com (A.C.D.-T.); thainasouza2@gmail.com (T.P.S.); keilabrainher@gmail.com (K.A.S.); ccfilg@yahoo.com.br (C.C.F.); ac_manhaes@yahoo.com.br (A.C.M.); 2Departamento de Ciências, Faculdade de Formação de Professores da Universidade do Estado do Rio de Janeiro, São Gonçalo 24435-005, RJ, Brazil; ribeiro_carvalho@yahoo.com.br

**Keywords:** psychiatric disorder, nicotine addiction, proteome, E-cigarette, mass spectrometry, prodrome, co-exposure, comorbidity

## Abstract

Tobacco misuse as a comorbidity of schizophrenia is frequently established during adolescence. However, comorbidity markers are still missing. Here, the method of label-free proteomics was used to identify deregulated proteins in the medial prefrontal cortex (prelimbic and infralimbic) of male and female mice modelled to schizophrenia with a history of nicotine exposure during adolescence. Phencyclidine (PCP), used to model schizophrenia (SCHZ), was combined with an established model of nicotine minipump infusions (NIC). The combined insults led to worse outcomes than each insult separately when considering the absolute number of deregulated proteins and that of exclusively deregulated ones. Partially shared Reactome pathways between sexes and between PCP, NIC and PCPNIC groups indicate functional overlaps. Distinctively, proteins differentially expressed exclusively in PCPNIC mice reveal unique effects associated with the comorbidity model. Interactome maps of these proteins identified sex-selective subnetworks, within which some proteins stood out: for females, peptidyl-prolyl cis-trans isomerase (Fkbp1a) and heat shock 70 kDa protein 1B (Hspa1b), both components of the oxidative stress subnetwork, and gamma-enolase (Eno2), a component of the energy metabolism subnetwork; and for males, amphiphysin (Amph), a component of the synaptic transmission subnetwork. These are proposed to be further investigated and validated as markers of the combined insult during adolescence.

## 1. Introduction

Schizophrenia (SCHZ) is a devastating psychiatric disorder that impacts not only patients but also their families and society. Its occurrence is associated with a broad range of adverse outcomes, including but not limited to cognitive decline, social isolation, stigma, unemployment and reduced life expectancy [1,2]. The available pharmacological approaches to treat this disorder are largely based on the manipulation of neurotransmitter system activities, and their partial efficacy has fueled the development and acceptance of the main hypotheses of SCHZ [3,4]. Corroborating these hypotheses, alterations in several neurotransmitter systems were identified in SCHZ patients [5] and in animal models [6]. However, the effectiveness of antipsychotics is limited [7], which suggests the involvement of other systems and pathways in the etiology and pathophysiology of this disorder. In this regard, recent studies showed significant changes in the expression of proteins not only related to synaptic signaling [8] but also involved with functions and pathways such as oxidative stress responses, the immune system, mitochondrial and peroxisomal function, the cytoskeleton, protein trafficking, energy/mitochondrial metabolism and clathrin-mediated endocytosis in SCHZ patients [9]. Corresponding findings were identified in animal models [10].

Two factors add extra layers of complexity to the identification of the mechanisms involved in this mental disorder. One of them is gender. In this regard, there is a higher prevalence and severity of negative and cognitive symptoms as well as worse response to treatment with antipsychotic medication in men, and the biological elements that underlie these differences are underinvestigated [11,12,13,14,15]. The second factor is the patients’ high vulnerability to substance use disorders, including nicotine addiction. The prevalence of conventional tobacco use in SCHZ patients is two to four times higher than the prevalence in individuals without a mental disorder, so that 50–90% of SCHZ patients smoke [16,17]. There was some speculation that nicotine potentiates psychosis [18], but some studies failed to corroborate these findings [19]. There is also evidence of procognitive effects of nicotine, improvement in the negative symptomatology [20] and mitigation of antipsychotic side effects [21]. Nevertheless, it is worth mentioning that smoking is a major factor that contributes to high mortality levels and reduced life expectancy in SCHZ patients [22]. The binding of nicotine to nicotinic cholinergic receptors (nAChRs) not only has a profound impact on the cholinergic system but also, indirectly, modifies the functioning of other neurotransmitter systems [23,24], several of which are involved in the SCHZ pathophysiology [25,26]. However, so far, the mechanisms that underlie nicotine effects on SCHZ symptomatology are not established. This could be due to the involvement of pathways other than those associated with the classic neurotransmitter systems. In support of this possibility, there is evidence that, in smokers and in animal models, nicotine affects not only synaptic transmission but also multiple pathways and biological processes, and several of those are also reported to be enriched in SCHZ proteomic studies. These include energy/mitochondrial metabolism, oxidative stress response, clathrin-mediated endocytosis and the cytoskeleton [27,28,29].

Frequently, attenuated positive, cognitive and negative symptoms of SCHZ are already observed before its diagnosis, including during adolescence [30,31,32]. These events coincide with experimentation and progression to misuse of both conventional combustible tobacco products [33] and the more recently available electronic nicotine delivery systems [34,35]. In accordance with these data, the prevalence of tobacco use in the prodromal phase of SCHZ was recently reported to vary from 17 to 46% [36] and from 59 to 78% in patients with first-episode psychosis when compared with healthy matched controls [37,38]. These findings indicate that the association between SCHZ and nicotine misuse is established early. They further point to adolescence as a critical period for both nicotine misuse and SCHZ and also as a window of opportunity for early intervention [39]. In this sense, the identification of pathways and biomarkers of this comorbidity’s early establishment could provide valuable information.

The medial prefrontal cortex (mPFC) is key to this investigation because it continues to mature during adolescence [40]. Also, this brain region plays roles in executive functions such as decision making, attention, working memory and cognitive flexibility [41] as well as social [42] and addictive behaviors [43], all of which are intimately associated with the symptomatology of SCHZ [2,44] and with mechanisms of nicotine addiction [45,46]. Despite that, there are scant studies in human subjects focused on the mPFC during this period of comorbid susceptibility. In addition to the limited access to brain samples, ethical concerns and difficulties in the control of confounding factors most likely have hampered advances in this area [47,48].

Although the association between SCHZ and nicotine misuse is frequent, smoking is seldom treated as a potential source of variation in epidemiological studies that use SCHZ subjects, and there is limited evaluation of its contribution to the outcomes. In this regard, the use of animal models may be of translational relevance. Phencyclidine, a noncompetitive N-methyl-D-aspartate (NMDA) receptor antagonist, is a dissociative drug that, in the period between the 1960s and the 1990s, was highly used as a drug of abuse under the street names of “Angel Dust” or “Peace Pill”. Interestingly, since its early use, phencyclidine has been described as a schizophrenomimetic drug because in healthy individuals, it triggers symptoms akin to those seen in SCHZ, while in SCHZ patients, the symptoms are worsened [49]. Nowadays, phencyclidine administration is a validated and widely used SCHZ model [50,51] as its repeated administration in adulthood produces behavioral alterations reminiscent of positive, negative and cognitive symptoms of this disorder [52]. This substance evokes deficits in prepulse inhibition [53] and neurochemical dysfunctions that correlate with those identified in SCHZ subjects [54]. These include hyperactivity in the mesolimbic dopaminergic system and hypoactivity in the mesocortical one, as well as hypofunction in the cortical glutamatergic and gamma aminobutyric acid (GABA) systems [50,52,54]. Most importantly, phencyclidine-induced SCHZ-like behavioral deficits were recently also demonstrated to occur when administration occurs during adolescence [55]. As for nicotine exposure, a subcutaneous osmotic minipump is an established model that has the advantage of being less stressful than other routes of administration, in addition to allowing for a constant dose and a controlled period of administration [56,57].

Proteomic approaches are useful for providing a broad view of the molecular alterations associated with SCHZ [9,58], which is also true for nicotine misuse [29]. However, much less is known regarding the comorbidity, particularly during adolescence. Here, we used a label-free quantitative proteomics approach to identify deregulated proteins in the mPFC of adolescent male and female mice modelled to SCHZ with and without a history of nicotine administration. With that, we aimed to preclinically delineate factors involved in the comorbidity during the early course of its establishment and to investigate whether there are relevant sex differences in these factors. We hypothesized that the analysis of the mPFC would lead to the identification of key pathological markers of the combined insult, which would correspond to exclusively deregulated proteins in mice modelled to SCHZ and exposed to nicotine.

## 2. Results

A total of 3594 unique peptides corresponding to 337 proteins were identified using a minimum of 2 peptides per protein that were present in at least two of the replicate analyses. A list of peptides and an alphabetized list containing identifiers for all mPFC proteins are available in Appendix A. An abundance ranking was also assigned to these proteins relative to their respective total independent spectra.

Considering the relevant sex differences in prevalence, symptom domains and severity described in SCHZ patients as well as animal models [59], for the analyses, mice were a priori separated by sex. Approximately 49.8% of proteins were differentially expressed only in one sex, and there were differences in the direction of effects (upregulation or downregulation) in males and females for the different comparisons described throughout the results section, which validated our approach (Appendix A). Also, for both males and females, changes in protein abundance depended on the drug(s) mice were exposed to, which corroborates the importance of analyzing independent factors (phencyclidine/nicotine). Firstly, PCP, NIC and PCPNIC were compared to CT mice with the aim of identifying the profile of deregulated proteins for each experimental group. Secondly, each group of mice exposed to phencyclidine (PCP and PCPNIC) was compared to the NIC group to isolate and subtract the effects of nicotine history per se. Thirdly, PCP and PCPNIC groups were compared to isolate proteins deregulated by phencyclidine and the combined insult. Next, to identify exclusive markers of PCP and NIC insults, the proteins commonly deregulated in all comparisons involving PCP (PCP vs. CT, PCP vs. NIC and PCP vs. PCPNIC) and NIC (NIC vs. CT, NIC vs. PCP and NIC vs. PCPNIC) were identified. Finally, exclusive markers of PCPNIC combined insults were also identified based on commonly deregulated proteins in all comparisons involving PCPNIC (PCPNIC vs. CT, PCPNIC vs. NIC and PCPNIC vs. PCP). For a summary of the findings, see Figure 1. To further characterize the relevance of the changes in the mPFC proteome of the model of SCHZ + nicotine comorbidity during adolescence, the exclusive markers of PCPNIC mice were plotted in interactome maps. This methodological approach was adapted from Barrera-Conde and collaborators [60].

### 2.1. Proteomic Profile of the mPFC of PCP, NIC and PCPNIC Mice When Compared to Controls

Figure 2A shows results from the proteomic analysis of the mPFC of PCP, NIC and PCPNIC mice when compared to CT. The PCPNIC group presented a higher number of deregulated proteins (males = 169, females = 156) when compared to PCP (males = 38, females = 36) and NIC (males = 134, females = 68) mice.

The analysis of PCPNIC mice also indicated a higher number of exclusively deregulated proteins (Figure 2A) in both males (n = 52) and females (n = 109) when compared to exclusive proteins identified in the PCP (males = 14, females = 8) and NIC (males = 23, females = 21) groups. Among the six more frequent Reactome pathways identified in PCPNIC mice, three (signal transduction, immune system and membrane trafficking) were affected in both males and females (Figure 2B). The innate immune system and signaling by Rho GTPases, Miro GTPases and RHOBTB3 were present among the other top Reactome pathways only in males. As for females, metabolism, metabolism of proteins and post-translational modification of proteins were enriched (Figure 2B). Detailed analyses of the impact of nicotine and phencyclidine isolated exposures are described in Appendix A.

In males, 16 proteins were commonly deregulated in PCP, NIC and PCPNIC when compared to CT mice. The analysis of females showed 14 proteins commonly deregulated (Figure 2C). The Reactome analysis indicated that none of the six more frequent pathways were shared between males and females. In males, those pathways were signal transduction, metabolism, immune and neuronal systems, transmission across chemical synapses and signaling by receptor tyrosine kinases. In females, transport of small molecules, post-translational modification and metabolism of proteins and other pathways related to mitotic cell division were the most frequent ones (Figure 2C).

A list of deregulated proteins involved in each comparison can be found in Appendix A.

### 2.2. Proteomic Profile of the mPFC of PCP or PCPNIC Mice When Compared to Nicotine-Exposed Ones

In total, in males, 114 proteins were differentially expressed in PCP and 59 proteins were differentially expressed in PCPNIC as compared to NIC mice. In females, 24 and 94 proteins were deregulated in PCP and PCPNIC, respectively (Figure 3A). Proteins that were not targeted by nicotine history were those exclusively deregulated in PCP and PCPNIC as compared to NIC mice. In males, the analysis indicated 84 proteins exclusively deregulated in PCP and 29 in PCPNIC mice. In females, those numbers were 12 for PCP mice and 82 for PCPNIC ones (Figure 3A).

Regarding the exclusively deregulated proteins in PCPNIC mice (Figure 3B), the functional characterization identified three out of the six top pathways enriched in both males and females (immune system and innate immune system and metabolism). When considering the other three top pathways, in males, developmental biology and hemostasis-related pathways stood out. In females, signal transduction, post-translational modification and metabolism of proteins were enriched.

The Reactome analysis of the exclusively deregulated proteins in PCP mice and a detailed list of the proteins involved in each comparison can be found in Appendix A and in Appendix A.

### 2.3. Proteomic Profile of the mPFC of PCPNIC Mice When Compared to PCP-Exposed Ones

To further characterize proteins differentially expressed by the combined insult, we compared PCPNIC to PCP mice. The data analysis unveiled 142 and 84 proteins deregulated in males and females, respectively. Among the six more frequent Reactome pathways, four (immune system, signal transduction, metabolism and metabolism of proteins) were affected in both males and females (Figure 3C), which indicates a high coincidence between sexes in response to the dual insult. The innate immune system and vesicle-mediated transport were present among the top six Reactome pathways only in males. As for females, membrane trafficking and post-translational protein modification aggregated more proteins (Figure 3C). A list of proteins involved in this comparison can be found in Appendix A.

### 2.4. Protein Markers in the mPFC of PCPNIC Mice

To identify specific protein markers of the SCHZ + nicotine comorbidity adolescent model, the proteins exclusively deregulated in the comparison between PCPNIC and CT (males = 52, females = 109) in PCPNIC when compared to NIC (males = 29, females = 82) and in PCPNIC when compared to PCP mice (males: n = 142, females: n = 84) were plotted in Venn diagrams. In males, 16 commonly deregulated proteins (protein markers of PCPNIC) were identified, while in females, there were 45 proteins (Figure 4A). The functional characterization of these proteins indicated that none of the six top pathways were common for males and females. In males, the top enriched pathways were innate immune system, developmental biology, axon guidance, nervous system development, platelet activation and MAPK family signaling cascades (Figure 4B). As for females, metabolism, post-translational protein modification and metabolism, signal transduction, immune system and adaptative immune system stood out (Figure 4B). The Reactome analysis of the protein markers of either PCP or nicotine exposure and a detailed list of the proteins involved can be found in Appendix A, and in Appendix A.

These data show a large number of protein markers of PCPNIC combined insults. To further understand the relevance of changes in the mPFC proteome of mice modeling the comorbidity during adolescence, we created interactome maps for males and females using these proteins (Figure 5). This approach allowed the identification of subnetworks. In order to facilitate the description and discussion of the results, each subnetwork was named in accordance with its main identified biological function (Table 1 and Table 2).

In males, the interactome map was composed of 12 PCPNIC protein markers (Figure 5A and Table 1). Ten proteins were grouped into four interconnected subnetworks named synaptic transmission, nervous system development, actin cytoskeleton and serine/threonine-protein phosphatases. As shown in Figure 5A, when considering individual proteins, amphiphysin (Amph), a component of the synaptic transmission subnetwork, stood out as centrally located and connected to the other subnetworks. A fifth subnetwork named energy metabolism was composed of two proteins and was isolated from the others.

As for the female interactome map, it was composed of 35 proteins (Figure 5B and Table 2). Thirty of them fit into five interconnected subnetworks enriched for oxidative stress, energy metabolism, serine/threonine-protein phosphatases, neurite outgrowth and G protein-coupled receptors signaling. The other four proteins comprised two small and isolated subnetworks named septin cytoskeleton and cell cycle. As shown in Figure 5B, peptidyl-prolyl cis-trans isomerase (Fkbp1a) and heat shock 70 kDa protein 1B (Hspa1b), both components of the oxidative stress subnetwork, and gamma-enolase (Eno2), a component of the energy metabolism subnetwork, stood out as centrally located. These three proteins were interconnected and linked to the serine/threonine-protein phosphatases and neurite outgrowth subnetworks.

Even though two subnetworks were shared between males and females (energy metabolism and serine/threonine-protein phosphatases), none of the proteins that formed the subnetworks were common to both sexes. In addition, all proteins from the male interactome map were downregulated when compared to CT, NIC and PCP groups, while for females, most proteins (76.5%) were upregulated (Table 1 and Table 2).

## 3. Discussion

The higher number of proteins differentially expressed in PCPNIC mice (PCPNIC vs. CT) when compared to the other groups (PCP vs. CT and NIC vs. CT) indicates that the combined insults lead to a worsened disbalance in the mPFC than each insult separately. These data are consistent with subthreshold effects of nicotine history and phencyclidine that, when combined, lead to disrupted expression of proteins. The higher number of potential protein markers of PCPNIC combined insults when compared to protein markers of SCHZ and NIC further corroborates this conclusion. Nevertheless, this study also identified a shared impact of phencyclidine and nicotine history, which supports evidence of interactive events between SCHZ and nicotine misuse in patients and comparable effects in animal models [55,61,62]. Consistent with this finding, despite the identification of proteins exclusively deregulated in the PCPNIC group, the top Reactome pathways were shared with other experimental groups. Thus, the common pathways indicate a functional overlap between SCHZ and nicotine history, while the exclusively deregulated proteins identify potential specific protein markers of PCPNIC combined insults within these pathways.

The proteins differentially expressed exclusively in PCPNIC mice and the interactome maps created with this database were sex-specific and corroborated previous evidence of sex differences in SCHZ patients [11,12,13,14,15]. While the interactome maps of both males and females were enriched for serine/threonine-protein phosphatases and energy metabolism, subnetworks specifically associated with nervous system development, synaptic transmission and actin cytoskeleton were present only in males. As for females, oxidative stress, neurite outgrowth, G protein-coupled receptors signaling, septin cytoskeleton and cell cycle aggregated more proteins. Overall, these pathways were previously shown to be compromised in SCHZ patients and, as discussed below, several of the potential specific protein markers of PCPNIC combined insults identified here were previously suggested to be markers of SCHZ. In this regard, it should be noted that studies on SCHZ patients often fail to report their smoking status and smoking history. Even when this information is provided, it is not uncommon for smoking to be overlooked as a potential contributing factor to the outcomes, and, as a result, SCHZ and SCHZ smoker patients are not segregated in the analysis. Similarly, frequently, either only one gender is included or gender is not treated as a separate factor in the analysis. These data suggest that several of the exclusively deregulated proteins identified in the PCPNIC group should be revisited and investigated as novel biomarker candidates for the comorbidity in males and/or females. Finally, we also identified new candidate markers as well as proteins that interconnected several subnetworks that should be considered in future studies on the subject. The ensuing paragraphs elaborate on these essential conclusions.

As the pathophysiology of both SCHZ and nicotine in the brain has multiple mechanisms, here, the discussion was focused on proteins that comprise interconnected subnetworks. For a discussion of isolated subnetworks identified in the male and female interactome maps, please refer to Appendix A.

### 3.1. Males mPFC Interactome Map of PCPNIC Mice

In adolescent males, the 12 protein markers of PCPNIC combined insults that comprised the interactome map were downregulated when compared to the CT, PCP and NIC groups. As discussed below, this consistent effect might be associated with the earlier onset and higher severity of symptoms identified in male patients diagnosed with SCHZ when compared to females [14,63]. Sex-biased effects were also identified in animal models of the disorder [63], even during adolescence [64].

In the last decades, the components of the synapse and the mechanisms involved in neurotransmission have become increasingly clear. Here, we found interconnected synaptic proteins as putative markers of the comorbidity. These proteins comprise the synaptic transmission subnetwork, which, in the interactome map of males, includes amphiphysin (*Amph*), synaptophysin (*Syp*), microtubule-associated protein 2 (*Map2*) and mitogen-activated protein kinase 3 (*Mapk3*). *Amph* expression was previously reported to be altered in SCHZ patients [65], and antiamphiphysin autoantibodies were associated with autoimmune psychosis [66,67]. *Amph* is involved in the invagination and fission steps of clathrin-mediated synaptic vesicle endocytosis [68] and, in the interactome map, it was directly linked to *Syp*, a protein localized in the membrane of synaptic vesicles implicated in the vesicular targeting of the plasma membrane and known to have reduced expression in several brain areas in SCHZ patients [69]. The altered expressions of *Amph* and *Syp* are consonant with the known disbalance of presynaptic neurotransmitter release mechanisms in SCHZ [69] and nicotine misuse [70,71]. However, other components of the synapse are also affected. In this regard, presynaptic dysregulation may negatively impact the structure and function of postsynaptic terminals. In accordance with this hypothesis, here, *Syp* was connected to *Map2*, whose reduction is a hallmark of SCHZ [72]. *Map2* is a core regulator of the microtubule network that contributes to cytoskeletal organization and dendritic arborization, both of which are essential for postsynaptic plasticity [72]. *Mapk3*, in turn, is a member of the MAP kinase family. MAP kinases, also known as extracellular signal-regulated kinases (ERKs), regulate cellular processes including proliferation and differentiation in response to extracellular signals. ERKs associate with microtubules and other cytoskeletal elements [73,74], which is consistent with their connection to *Map2* in the network. This connection may facilitate MAPK activity and lead to cytoskeletal remodeling [73,74]. A transcriptome-wide association study (TWAS) in a large SCHZ genomewide association study (GWAS) data set showed that *Mapk3* gene expression was genetically correlated with SCHZ [75]. MAPK signaling was also associated with nicotine dependence phenotypes in a recent GWAS study [76].

*Amph* was situated in a central position in the male map, being connected to the actin cytoskeleton subnetwork via actin-related protein 3B (*Actr3b*), which, in turn, was linked to adenylyl cyclase-associated protein 1 (*Cap1*). Actin microfilaments are components of the cytoskeleton [77], which, in neurons, is responsible for the generation, maintenance and remodeling of axons and dendritic arbors. The concept that a dysfunctional cytoskeleton underlies dendritic architecture and synaptic connectivity changes observed in the brains of SCHZ patients is supported by evidence from clinical studies and animal models [72]. Regarding nicotine, countless pieces of evidence of synaptic remodeling and its association with addiction mechanisms also point to the cytoskeleton as a target [78,79,80,81] and suggest interactive events between these diseases. Here, the identification of the actin cytoskeleton subnetwork indicates that the cytoskeleton is vulnerable to the combination of the insults. *Actr3b* is a subunit of the actin-related protein 2/3 complex, a major component of the actin cytoskeleton that serves as a nucleation site for new actin filaments and actin filament polymerization in functions such as phagocytosis and vesicular trafficking [82]. In neurons, the actin-related protein 2/3 complex is also involved in dendritic spine morphogenesis, contributing to the maturation of the dendritic spine phenotype [83]. Specifically, *Arpc3* has been associated with developmental and psychiatric disorders and discovered to be an interactor of SCHZ genes [84]. It was also shown to coimmunoprecipitate with  β2*-nAChRs from the human postmortem temporal cortex [85] and to be indirectly modulated by nicotine [86]. As for *Cap1*, it regulates actin dynamics and cytoskeleton remodeling processes [87]. Its expression is altered in the neonatal ventral-hippocampal lesion rat model of SCHZ [88].

Both the synaptic transmission and the actin cytoskeleton were interconnected and connected to the serine/threonine-protein phosphatases subnetwork, and *Amph* is the protein that intermediated these connections. In the serine/threonine-protein phosphatases subnetwork, serine/threonine-protein phosphatase 2B catalytic subunit alpha (*Ppp3ca*) and serine/threonine-protein phosphatase PP1-beta catalytic subunit (*Ppp1cb*) were deregulated. *Ppp3ca* is the catalytic subunit of calcineurin. Among phosphoprotein phosphatases (PPPs), calcineurin is one of the most abundant in the central nervous system, acting on multiple substrates. Calcineurin dephosphorylates AMPA receptors, the most relevant mediator of fast synaptic transmission at excitatory synapses. AMPA receptor dephosphorylation triggers their internalization and decreases synaptic efficacy, contributing to long-term depression, which involves modifications in the cytoskeleton [89,90]. Calcineurin was also proposed to be a regulator of bidirectional plasticity due to its ability to dephosphorylate GABAA receptors, an effect that may reduce inhibitory transmission and lead to the disinhibition of excitatory neurotransmission [89]. As for *Ppp1cb*, it is one of the three catalytic subunits of protein phosphatase 1 (PP1). PP1 is also involved in neuroplasticity events, including long-term depression [91]. PP1 inhibition leads to the potentiation of dopaminergic signaling, which underlies both SCHZ and addiction mechanisms [92].

In addition to the associations described above, *Amph* was connected to dihydropyrimidinase-related protein 5 (*Dpysl5*), a protein that, together with dihydropyrimidinase-related protein 1 (*Crmp1*), comprises the nervous system development subnetwork. Both *Dpysl5* and *Crmp1* are developmentally regulated proteins highly expressed throughout brain development when they are involved in processes such as axon outgrowth, neuronal migration, signal transduction and neuronal differentiation [93]. In adulthood, their expression is more restricted, as they are prominent in areas involved in neuroplasticity [93]. Accordingly, the distinct roles of these proteins depends on the developmental stage of the organism, which suggests a dynamic involvement in the structural and functional synaptic alterations that occur in SCHZ and nicotine misuse. Altered expression of *Crmp1* was identified in the cingulate cortex of SCHZ patients [94]; however, *Dpysl5* is a novel comorbidity candidate, since there are no previous reports of its dysregulation in preclinical or clinical studies on SCHZ and nicotine/tobacco. Together, data from the male interactome network corroborate previous findings that associate SCHZ and nicotine exposure with severe synaptic disbalance and indicate that several proteins with distinct (direct and indirect) roles in neurotransmission contribute to the effects of PCPNIC combined insults in the model of SCHZ + nicotine comorbidity in adolescence used in this study.

It should be noted that males exhibited significantly fewer exclusive protein markers of PCPNIC combined insults than females, which could indicate that the expression of the comorbidity in males largely reflects the combined effects of each disease. If similar results occur in SCHZ patients who smoke, this suggests that there are common contributions of each disease to the pathophysiology of SCHZ + NIC misuse comorbidity, which could guide future studies that aim to identify therapeutic targets of each disease and their comorbid occurrence.

### 3.2. Female mPFC Interactome Map of PCPNIC Mice

In females, 26 out of the 35 protein markers of PCPNIC combined insults that comprised the interactome map were upregulated when compared to the CT, PCP and NIC groups. This is a very distinct pattern from that identified in males, in which all proteins were downregulated. Considering previous evidence of sex differences in animal models of SCHZ and in SCHZ patients [14,63,64], it is possible that the pattern identified in females is related to a tentative compensatory effect. In case similar events occur in patients, those could play a role in the less severe negative and cognitive symptomatology and course of the disorder described in SCHZ women [95].

The oxidative stress was the largest subnetwork identified in the current study, composed of 10 proteins. Two components of this subnetwork, heat shock 70 kDa protein 1B (*Hspa1b*) and peptidyl-prolyl cis-trans isomerase (*Fkbp1a*), were situated in a central position in the female interactome map and were connected to three other subnetworks. Even though the current set of data does not allow the identification of causal relations, the associations detailed below point to *Hspa1b* and *Fkbp1a* as strong biomarker candidates for PCPNIC combined insults in females, both of them being critical for proteostasis.

*Fkbp1a* is an FK506-binding protein (FKBPs) under the immunophilin family, a division of the peptidyl-proline isomerases (PPIases) chaperone superfamily. By catalyzing the cis/trans isomerization of proline, PPIases act as a regulatory switch during folding, activation and degradation of a wide variety of proteins [96]. FKBPs are also involved in the modulation of gene transcription, protein trafficking and cellular signaling [97]. In the female interactome map, *Fkbp1a* was upregulated when compared to CT, PCP and NIC groups, and it constitutes a novel comorbidity candidate. Its deficiency has been associated with an enhancement in long-lasting hippocampal long-term potentiation, memory and perseverative behaviors [98]; however, future studies on the impact of its increased expression are warranted.

Heat shock proteins act as molecular chaperones and demonstrate crucial protective functions in stressed cells. *Hspa1b* is a member of the 70 kDa heat shock protein family (Hsp70), which is involved with virtually all stages in the life of proteins from synthesis to degradation [99,100]. *Hspa1b* was repeatedly associated with SCHZ [101,102,103,104]. *Hspa1b* rs1061581 polymorphism was shown to contribute to susceptibility to paranoid SCHZ and clinical presentation of the disease in a Polish subpopulation [104]. *Hspa1b* haplotypes are associated with improved positive and negative symptoms in inpatients [105] and, recently, *Hspa1b* s1061581 genotypes were associated with improved treatment responses in first-episode psychosis [106]. *Hspa1b* was one of the protein markers that were downregulated in PCPNIC female mice when compared to the other experimental groups, which further indicates that it contributes to the pathophysiology of SCHZ.

*Fkbp1a* and *Hspa1b* were interconnected, and both of them were directly linked to heat shock-related 70kDa protein 2 (*Hspa2*), which is consistent with these proteins’ roles in cell proteostasis. When considering the oxidative stress subnetwork, *Hspa1b* held most of the connections. Interestingly, while *Hspa1b* was downregulated in PCPNIC females when compared to CT, NIC and PCP mice, all other proteins connected to it were upregulated, which raises the possibility that *Hspa1b* played a role in this outcome. *Hspa1b* was directly linked to protein disulfide-isomerase A3 (*Pdia3*), Parkinson disease protein 7 (*Park7*), and *peroxiredoxin-1* (*Prdx1*). Park7 has been described as a redox sensor. In the event of oxidative stress, it is oxidized and translocated to the mitochondria [107,108]. *Park7* was also connected to *Pdia3*, a rapid-response vitamin D receptor at the cell membrane, and to *Prdx1*, which, in turn, was connected to glutathione S-transferase Mu 1 (*Gstm1*), and glutathione S-transferase P 1 (*Gstp1*), all of them contributors to the central nervous system redox homeostasis [109]. The deregulation of the above-mentioned proteins is consistent with evidence for the role of redox imbalance in the etiology and pathogenesis of SCHZ [110] and nicotine misuse [111,112]. This finding also lends support to the hypothesis of glutamatergic hypofunction in SCHZ, as there is increasing evidence that NMDA hypofunction, characteristic of SCHZ, and redox imbalance are reciprocally linked: the redox balance controls NMDA receptor activity, and NMDA receptor hypofunction leads to oxidative stress [113].

The oxidative stress and the serine/threonine-protein phosphatases subnetworks were interconnected. In the latter, one catalytic subunit of PP1 (serine/threonine-protein phosphatase PP1-gamma catalytic subunit (*Ppp1cc*)) and a catalytic and a regulatory subunit of PP2A (serine/threonine-protein phosphatase 2A catalytic subunit beta isoform (*Ppp2cb*) and serine/threonine-protein phosphatase 2A 65 kDa regulatory subunit A alpha isoform (*Ppp2r1a*)) were upregulated when compared to the other experimental groups. In the male interactome map, PP1 was also affected; however, a different catalytic subunit was targeted, and it was downregulated. As previously mentioned, PP1 inhibition prevents long-term depression [91]. In case PP1 upregulation has the opposite effect, in females, it could contribute to the frontal cortex hypofrontality identified in SCHZ patients [114]. As for PPA2, it is a D2R partner [115], and DR2 stimulation triggers a PPA2-mediated cascade of events that ultimately leads to altered neurodevelopment and cognition associated with SCHZ [116,117]. *Fkbp1a* and *Hspa1b* were the proteins of the oxidative stress subnetwork that mediated the connection with PP1 and PP2A; however, future studies are needed to investigate the functional association between these proteins and the deregulated components of the serine/threonine-protein phosphatases subnetwork. *Isoform 2 of Liprin-alpha-3* (*Ppfia3*) was the fourth component of the serine/threonine-protein phosphatases subnetwork, connected to both PP2A deregulated proteins in the map. *Ppfia3* is a member of the LAR protein tyrosine phosphatase-interacting protein (liprin) family. Liprins interact with serine/threonine-protein phosphatases among other proteins [118]; however, there is no previous evidence of direct interaction between *Liprin-alpha-3* and *PPA2A*. Similarly, to the best of our knowledge, this is the first report of *Ppfia3* association with SCHZ and nicotine exposure; however, liprins-alpha are known to play roles in synapse development and maintenance [119], which suggests that this protein could be of interest as a candidate marker of the comorbidity. Liprins-alpha are enriched at both pre- and postsynaptic sites [119]. They are components of the presynaptic cytomatrix at the active zone and are important for the organization of synaptic vesicles [120]. At postsynaptic sites, liprins-alpha contribute to dendritic clustering and surface expression of AMPA receptors [121,122]. Liprin-alpha involvement in synaptic neuroplasticity is further demonstrated by evidence that interactions between *Ppfia3* and AMPA receptors are required for muscarinic receptor-mediated long-term depression [123].

The energy metabolism was the second largest subnetwork identified in the current study, composed of eight proteins. These proteins are involved in the glycolytic cycle (glucose-6-phosphate isomerase (*Gp1*), glyceraldehyde-3-phosphate dehydrogenase (*Gapdhs*), gamma-enolase (*Eno2*) and pyruvate kinase (*Pklr*)), the citric acid cycle (citrate synthase_mitochondrial (*Cs*) and Acetyl-CoA acetyltransferase_mitochondrial (*Acat1*)), the malate–aspartate shuttle (aspartate aminotransferase_mitochondrial (*Got2*)) and the glutamate–glutamine cycle (glutamine synthetase (*Glul*)). Interestingly, while most proteins from the glycolytic cycle were upregulated, *Cs*, which catalyzes the first step in the citric acid cycle, and *Acat1*, which breaks down fatty acids into acetyl-CoA, were downregulated in the PCPNIC when compared to the other groups. These findings may indicate an increased rate of glycolysis; however, they also suggest that pyruvate, the final product of glycolysis, is diverted from the citric acid cycle. Altogether, while these data are consistent with evidence for the role of aberrant energy metabolism in the etiology and pathogenesis of SCHZ [124,125], they also indicate that nicotine may play a role in the outcomes.

Of note, *Glul*, a component of the glutamate–glutamine cycle that catalyzes the amidation of glutamate to glutamine in astrocytes, was upregulated, corroborating postmortem studies that showed increased *Glul* in the prefrontal cortex of SCHZ patients [126]. The product of the reaction, glutamine, is released in the extracellular space and reaccumulates into neurons, where it is used as a glutamate precursor. *Glul* is an oxidative-sensitive enzyme [127]; therefore, the oxidative disbalance identified in PCPNIC mice may be in part responsible for *Glul* increased levels and is consistent with its connection to the oxidative stress subnetwork in the interactome map. Furthermore, in addition to glutamine, alpha-ketoglutarate from the citric acid cycle can function as a precursor for glutamate; therefore, a disbalance in glycolysis and the citric acid cycle may impact glutamate metabolism. Since glutamate is a precursor for GABA, this latter neurotransmitter may also be affected [128,129]. Accordingly, the energy metabolism disbalance may play a role in the increased expression of *Glul*. *Glul* is reported to be neuroprotective by inactivating glutamate and preventing its neurotoxic effects [130]. Since estradiol is known to increase *Glul* mRNA and protein expression [130], whether *Glul* neuroprotective mechanisms contribute to a less severe symptomatology of SCHZ in women smokers is of interest for future investigation. Of note, in the interactome map, *Glul* was directly linked to *Got2*, a protein expressed in the mitochondria, where it converts glutamate into aspartate and alpha-ketoglutarate [131]. Accordingly, *Got2* upregulation suggests that it contributes to glutamatergic hypofunction in PCPNIC females.

*Eno2*, also a component of the energy metabolism subnetwork, was associated with SCHZ, both in first-episode patients [132] and in treatment-refractory SCHZ [133]. In the female interactome map, several proteins from the oxidative stress and neurite outgrowth subnetworks converge to *Eno2*, suggesting a central role of this protein in the pathophysiology of PCPNIC females. Even though both *Fkbp1a* and *Eno2* were shown to be deregulated in response to insults [134,135], connections between *Eno2* and other oxidative stress proteins (*Hspa1b* and to *Park7*, *Atp6v1a* and *Gstp1*) in the interactome map need further investigation. *Eno2* is responsible for the reversible conversion of D-2-phosphoglycerate and phosphoenolpyruvate in glycolysis and gluconeogenesis. Furthermore, it has neurotrophic and neuroprotective effects on cultured neocortical neurons [136]; therefore, its upregulation in PCPNIC females suggests that it might contribute to the less severe symptomatology of SCHZ in women. This role is supported by findings of *Eno2* connection to Thy-1 membrane glycoprotein (*Thy1*) in the neurite outgrowth subnetwork. *Thy1* is a small glycoprotein anchored to the outer leaflet of the plasma membrane. It has an inhibitory effect on neurite outgrowth [137]; therefore, its downregulation in the female interactome map suggests a protective role. Despite that, *Thy1* knockout shows deficient social behavior and impaired long-term potentiation formation due to exacerbated GABAergic inhibitory activity in the dentate gyrus [137]. *Eno2* connections to transgelin-3 (*Tagln3*) and neuromodulin (*Gap43*) further suggest a deleterious outcome. *Tagln3*, an F-actin binding protein, regulates neurite outgrowth [138]. In the female map, *Tagln3* is connected to *drebrin* (*Dbn1*), also an F-acting binding protein and highly expressed in dendritic spines, where its accumulation level is regulated by NMDA-mediated synaptic activity [139]. Synaptic NMDA receptor activation reduces *Dbn1* levels in dendritic spines, which facilitates F-actin polymerization and long-term potentiation. Accordingly, our finding of upregulated *Dbn1* in PCPNIC females is consistent with the hypofrontality theory of SCHZ and reduced synaptic plasticity. As for *Gap43*, it has a supporting role in the regulation of presynaptic vesicular function, axonal growth and synaptic remodeling associated with neuroplasticity [140]. *Gap43* is directly linked to myristoylated alanine-rich C-kinase substrate (*Marcks*) in the interactome map. Both *Gap43* and *Marcks* are members of the family of neuronal growth-associated proteins and have a number of properties in common [141], possibly cooperating with common physiological effects. Both regulate actin cytoskeleton dynamics, neurite outgrowth and ultimately, memory and learning [142,143]. *Gap43* and *Marcks* were downregulated when compared to all other experimental groups, which suggests that they contribute to the pathogenesis of the comorbidity. In accordance with these findings, *Gap43* [144] and *Dbn1* [145] mRNA and *Marcks* protein levels [146,147] were previously shown to be deregulated in the prefrontal cortex of SCHZ patients.

Evidence of aberrant intracellular signaling in SCHZ comes from diverse lines of research that suggest a critical role of G protein-coupled receptors and their downstream pathways in the disease development, progression and treatment [148,149,150]. As for nicotine, it is well-known that it indirectly interferes with G protein-coupled receptors and their signaling cascades, impacting neurite outgrowth and memory [151,152], which are known to be affected in SCHZ. Consistent with these data, in the female interactome map, three upregulated proteins comprised the G protein-coupled receptors signaling subnetwork as follows: guanine nucleotide-binding protein G(olf) subunit alpha (*Gnal*), guanine nucleotide-binding protein subunit alpha-12 (*Gna12*) and guanine nucleotide-binding protein G(I)/G(S)/G(T) subunit beta-2 (*Gnb2*). *Gnal* is coupled with the dopamine D1 receptor and activates adenylyl cyclase [153]. Accordingly, *Gnal*^+/−^ knockout rats show phenotypes associated with deficient dopaminergic transmission, reduced *Arc* expression, increased surface expression of AMPA receptors and loss of D2 receptor-mediated corticostriatal long-term depression [154]. Distinctively, *Gna12* was reported to bind to several receptors, including the 5HT2c, α1-adrenoceptor and muscarinic M1 cholinergic ones [155]. *Gna12* often uses Rho as an intermediate, which leads to changes in the actin cytoskeleton [155]. It has several proximal effectors, and one of particular interest is the glutamate transporter EAAT4-associated protein (GTRAP48) [155], which, by interacting with the intracellular domain of EAAT4, modulates glutamate transport activity in neurons [156], possibly impacting neuronal excitability. Recently, a study that investigated genomic factors that regulate gray matter density in healthy subjects highlighted *Gna12* as a major player. In frontal and parietal cortical regions, *Gna12* was significantly enriched in pathways including long-term depression, CREB signaling in neurons and axonal guidance [157]. Interestingly, a combinatory analysis of multiple GWAS identified the *Gna12* gene with a joint effect significantly associated with SCHZ. Also, the neuronal-related processes—long-term depression and CREB signaling [158]—overlapped those that were identified as involved in the regulation of gray matter density [157]. Finally, the use of machine learning methods identified *Gnb2*, the third protein of the subnetwork, in multiple SCHZ-associated pathways in the prefrontal cortex, including the glutamatergic and the dopaminergic [159]. In fact, all three genes [158,159,160] are considered candidate SCHZ-associated genes.

## 4. Materials and Methods

### 4.1. Materials

Ketamine (Ceva, Paulínia, SP, Brazil), xylazine (Syntec, Tamboré, SP, Brazil), flunixin (Ucb/Vet, Jaboticabal, SP, Brazil) and enrofloxacin (Venco, Londrina, PR, Brazil) are specifically for veterinary use. Nicotine-free base, dithiothreitol (DTT), iodoacetamide (IAA), formic acid, sodium deoxycholate and triethylammonium bicarbonate buffer (TEAB) were purchased from Sigma Chemical Co. (St. Louis, MO, USA). Phencyclidine was purchased from Alomone Labs (Jerusalem, Israel). Osmotic minipumps came from Alzet (Cupertino, CA, USA). Promega Biotecnologia do Brasil (São Paulo, SP, Brazil) was the source of trypsin, and Thermo Fischer Scientific (Waltham, MA, USA) was the source of the resin column model POROS 20 R2 and Qubit protein assay kit. The nanoEase Symmetry C18 TRAP column (5 µm, 180 µm × 20 mm), the nanoAcquity BEH130 C18 reversed-phase column (1.7 µm, 75 µm × 100 mm), [Glu1]-Fibrinopeptide B human (GFP) and Total Recovery vials were purchased from Waters Corp (Milford, MA, USA). Acetonitrile came from Merck HGaA (Darmstadt, Germany).

### 4.2. Animals and Treatment

All experimental procedures were approved by the Institute of Biology/UERJ Ethical Committee for Animal Research (protocol#: 033/2018), minimizing the number of animals used and avoiding animal suffering, in accordance with Brazilian Law # 11.794/2008. All mice were kept in our animal facility and housed in groups of 2–5 at 21–22 °C on a 12 h light/dark cycle (lights on at 1:00 a.m.). Access to food and filtered water was *ad libitum*. Animals were derived from a C57BL/6 colony maintained at the Universidade Federal Fluminense (Niteroi, RJ, Brazil) for over 60 generations.

As shown in Figure 6, from postnatal (PN) 37 to PN44, mice were exposed to nicotine. Briefly, at PN37, mice were anesthetized with xilasine (20 mg/kg, i.p.) and ketamine (100 mg/kg, i.p.), a small area on the back was shaved, and an incision was made to permit s.c. insertion of an osmotic minipump (model 1007d). Minipumps were prepared on the day preceding implantation with nicotine-free base diluted in milli-Q water (pH 6.0) to deliver an initial dose rate of 24 mg/kg of nicotine per day. Incisions were closed after minipump insertion and mice were given s.c. injections of flunixin (2.5 mg/kg) and enrofloxacin (2.5 mg/kg) for pain management and infection prevention both right after closure and on the following day. The period of exposure to nicotine was intended to parallel human exposure during adolescence, a key period for initiation of tobacco and electronic nicotine delivery systems use [34,161,162]. The dose rate used in the current study resulted in nicotine serum levels [56] that were within the range of those found in smokers [163,164]. Control mice were implanted with minipumps containing milli-Q water. Animals were allowed to recover from surgery in their home cages.

From PN38 to PN52, the animals were treated with daily injections of phencyclidine, an NMDA receptor antagonist used to model SCHZ, at a dose of 2.5 mg/kg (s.c.) [59,165]. On the last day of phencyclidine treatment (PN53), the dose used was 10 mg/kg (s.c.) [166]. The period of phencyclidine treatment was chosen based on evidence that the prodromal stage of the disorder begins early, still during adolescence [30,31,32] and that, in most cases, the first psychotic event in SCHZ occurs between late adolescence and early adulthood [31,167]. Control mice were injected with a saline solution (NaCl 0.9%). Accordingly, mice were distributed into four experimental groups: CT (control group), PCP (phencyclidine-exposed group), NIC (nicotine-exposed group) and PCPNIC (phencyclidine- and nicotine-exposed group).

At PN53, after the end of nicotine exposure but still during the period of phencyclidine treatment, mice were decapitated, and the brains were frozen in liquid nitrogen and stored at −76 °C. The 9-day interval between the end of nicotine exposure and sacrifice aimed to assess lasting effects of nicotine without the presence of confounding factors associated with acute withdrawal signs [168,169,170]. Samples were punched from the frozen brains using a cryostat (Hyrax C25, Zeiss, Jena, Germany) and a cylindric brain puncher (internal diameter 1.0 mm). The length and location of the punches followed the coordinates described in the Paxinos and Franklin stereotaxic atlas [171]: 0.6 mm of length (starting at 2.1 mm anterior to Bregma), aiming at the infralimbic and prelimbic areas. The samples were frozen at −76 °C for posterior proteomics analysis.

### 4.3. Protein Extraction

In order to guarantee minimum sample concentration for the analysis, pools of samples from different mice were used instead of individual samples. Punches of the medial prefrontal cortex from two animals within each experimental condition and sex were mixed to produce one sample pool. Four sample pools of each experimental group and sex were used for a total of eight different animals in each experimental condition and sex, resulting in four biological replicates and three technical replicates. Seventy milligrams of each sample pool were extracted with 500 μL of buffer (50 mM TEAB pH 8.5, 0.1% sodium deoxycholate). The samples were briefly vortexed and then incubated in an ultrasonic bath (SolidSteet, Piracicaba, SP, Brazil) for 30 min at 6 °C. After centrifugation at 18,400× *g* for 20 min at 4 °C (Eppendorf, centrifuge 5427R, Hamburg, Germany), the supernatants were transferred to new identified tubes. The extraction was performed twice, and the respective supernatants were merged. Acetonitrile with 0.1% formic acid was added to the supernatant for protein precipitation in a 4:1 proportion (*v*/*v*), centrifuged at 18,400× *g* for and 4 °C for 10 min and washed twice with cold acetone. Proteins were resuspended in TEAB 50 mM pH 8.5, dosed using a Invitrogen Qubit® fluorometer (Thermo Fischer Scientific, USA) and processed as described previously [172].

### 4.4. Protein Digestion for NanoULPC-MSE Analysis

Proteins were treated with 0.01% of sodium deoxycholate, reduced by the addition of 10 mM DTT at 56 °C for 30 min and alkylated with IAA 20 mM at 25 °C for 30 min. Afterward, samples were digested with sequencing-grade trypsin in a 1:100 trypsin:protein ratio in TEAB 50 mM, pH 8.5 for 18 h at 37 °C under agitation on a Thermomixer Comfort (Thermo Fischer Scientific, USA). Peptides were desalinated and concentrated through reverse-phase chromatography with a resin column POROS 20 R2 and eluted with 70% acetonitrile/0.05% formic acid. The peptides were dried, resuspended in 0.1% formic acid and quantified (using a Qubit protein assay kit), and the supernatants were transferred to Waters Total Recovery vials. Four biological replicates were analyzed (three technical replicates).

### 4.5. NanoUPLC-MSE Analysis

The nanoUPLC analysis of tryptic peptides was carried out using a nanoACQUITY UPLC system (Waters Corp., Milford, MA, USA) coupled to a Synapt G2-Si high-definition mass spectrometer (HDMS) (Waters Corp., Manchester, UK). The chromatographic system was equipped with a nanoEase Symmetry C18 (5 µm, 180 µm × 20 mm) TRAP column and a nanoAcquity BEH130 C18 (1.7 µm, 75 µm × 100 mm) reversed-phase column. Mobile phase A consisted of water and mobile phase B consisted of acetonitrile, both with 0.1% (*v*/*v*) formic acid. The peptides were separated using MSE chromatographic methods with the following gradient: from 3 to 50% of B in 60 min; then a cleaning column gradient of 50 to 85% of B for 1 min and maintained in 85% of B for 5 min; then from 85 to 3% of B in 1 min. The flow rate was 450 ηL·min^−1^. The analytical column temperature was maintained at 35 °C, and the sample manager temperature was 9 °C. The lock mass was derived from the auxiliary pump using a constant flow rate of 500 ηL·min^−1^ at a concentration of 200 fmol of GFP to the reference sprayer of the NanoLockSpray source.

The mass spectrometer was operated in the resolution mode for all measurements. All analyses were carried out using nano-electrospray ionization in the positive ion mode (nanoESI+) and a NanoLockSpray (Waters, Manchester, UK) ionization source. The lock mass channel was sampled at a frequency of 30 s. The ion source block temperature was set to 100 °C, and capillary voltage was set to 3 kV. The time-of-flight analyzer of the mass spectrometer was calibrated with an MS/MS spectrum of GFP solution. The final instrument calibration was obtained using the signal of GFP double-charged precursor ion [M + 2H]^2+^ = 785.8426. The exact mass retention time (EMRT) from multiplexed DIA scanning (MSE) MS analyses was collected in an alternating low-energy and elevated-energy acquisition mode. In the elevated collision energy mode, the collision energy was increased from 15 to 55 eV applied to the trap “T-wave” CID (collision-induced dissociation) cell with argon gas. The radiofrequency was adjusted such that the ions were effectively acquired from m/z 50 to 2000, which ensured that any masses of less than 50 m/z observed in the elevated-energy spectra were only derived from dissociations in the collision cell.

To minimize methodological variations that might impact the results, during all runs, the analytical column and the sample manager temperatures were monitored constantly and maintained stable. The chromatographic column condition was also monitored every 12 sample runs.

### 4.6. Data Processing

In order to identify and quantify peptides and proteins, the MS data were processed and searched using the Progenesis QI for Proteomics version 4.2 (Nonlinear Dynamics, USA). The UNIPROT protein databank release 2022-03 (http://www.uniprot.org, accessed on 23 March 2022) with specific annotations for *Mus musculus* was utilized. A reverse database was generated for monitoring the false-positive rate (false discovery rate—FDR). The parameters for database searching were tryptic peptides with a maximum of two missed cleavages, maximum protein mass of 650 kDa, carbamidomethylation of cysteine as fixed modification and oxidation of methionine as variable modifications. The parameters set as default were peptide mass error tolerance of 10 ppm, fragment mass error tolerance of 20 ppm and maximum of 1% of the FDR. At the protein level, the criteria included the detection of at least two fragment ions per peptide, three or more fragments per protein and the determination of at least two peptides per protein. For protein identification, the exact mass applied was less than 20 ppm and the score was more than 10. Relative quantification was determined from the absolute intensities obtained and considered using the Hi3 (Top3)-based quantitation method, as described in the literature [173]. The samples were compared in pairs to obtain expression data (CT, PCP, NIC and PCPNIC groups) and the following filters were used: coefficient of variance (maximum CV of 30%), maximum fold change (>1.5) and ANOVA (*p* < 0.05). Only coexisting proteins in both conditions and proteins that were present in the 3 replicates (3/4) were considered to compare the conditions.

Gene ontology annotation including biological processes and Reactome pathways was performed using the online Protein Analysis Through Evolutionary Relationships (PANTHER) classification system (http://www.pantherdb.org, accessed on 25 May 2022) [174]. The jvenn tool online (http://jvenn.toulouse.inra.fr/app/index.html, accessed on 17 August 2022) was used to draw Venn diagrams [175] to visualize all possible intersections among proteome datasets (https://bioinfogp.cnb.csic.es/tools/venny_old/index.html, accessed on 17 August 2022). Protein–protein interaction analysis was performed using STRING (https://string-db.org/, accessed on 13 July 2022) [176] with the minimum confidence level accepted for interactions set to medium (0.4). Data analysis was performed by an experimenter blind to the experimental groups.

The mass spectrometry proteomics data were deposited in the ProteomeXchange Consortium via the PRoteomics Identification Database (PRIDE) [1] partner repository (http://www.ebi.ac.uk/pride, accessed on 22 March 2022) with the dataset identifier PXD043626 (username: reviewer_pxd043626@ebi.ac.uk, password: mfmtV1DL).

## 5. Conclusions

The methodological approach used in the current study allowed the identification of potential pathological markers of combined phencyclidine and nicotine exposure during adolescence, which corresponded to exclusively deregulated proteins in the mPFC of male and female mice. Due to marked sex-selective effects, the stronger comorbidity biomarker candidates were distinct in males (Amph) and females (Fkbp1a, Hspa1 and Eno2), which is consistent with the recommendation that future studies consider gender/sex when interpreting the results. There were few new comorbidity candidates (e.g., Dpysl5 for males, Fkbp1a and Ppfia3 for females); these were the proteins for which there were no previous reports of dysregulation in preclinical or clinical studies on SCHZ and nicotine/tobacco. However, considering that the impact of smoking on SCHZ has frequently been overlooked in clinical studies, our results raise the possibility that SCHZ candidates reported in previous studies are in fact biomarker candidates for SCHZ + nicotine-misuse comorbidity.

A limitation of the current study is that the data have not yet been complementarily validated, which can be conducted either by using techniques such as immunoblot or through biological replication using independent samples. Accordingly, future studies to confirm the obtained results using suitable cellular models and molecular tools aimed at clarifying the mechanisms associated with PCPNIC combined insults are warranted.

The subnetworks identified here covered a wide range of pathways associated with SCHZ, and those extended beyond the neurotransmitter systems pathways classically associated with this disorder. Despite that, frequently, the interconnected proteins that comprised these subnetworks are known to play roles in modulating components of neurotransmitter systems such as the glutamatergic one; closing a loop and reinforcing the neurotransmitter systems disbalance is central to SCHZ and, importantly, also to the comorbidity. Finally, all deregulated proteins were identified during adolescence in mice, a period that, in humans, is marked by both tobacco initiation and the emergence of SCHZ symptoms. Despite the recognition of adolescence as critical for the establishment of the comorbidity, few studies in animal models of SCHZ and nicotine exposure investigated interactions between these conditions during this period of development. In this sense, the current pathfinder study has the potential to direct future studies aiming to identify mechanisms of nicotine interference in the early course of SCHZ and, possibly, to create more efficient treatments tailored for early intervention.

## Figures and Tables

**Figure 1 ijms-24-14634-f001:**
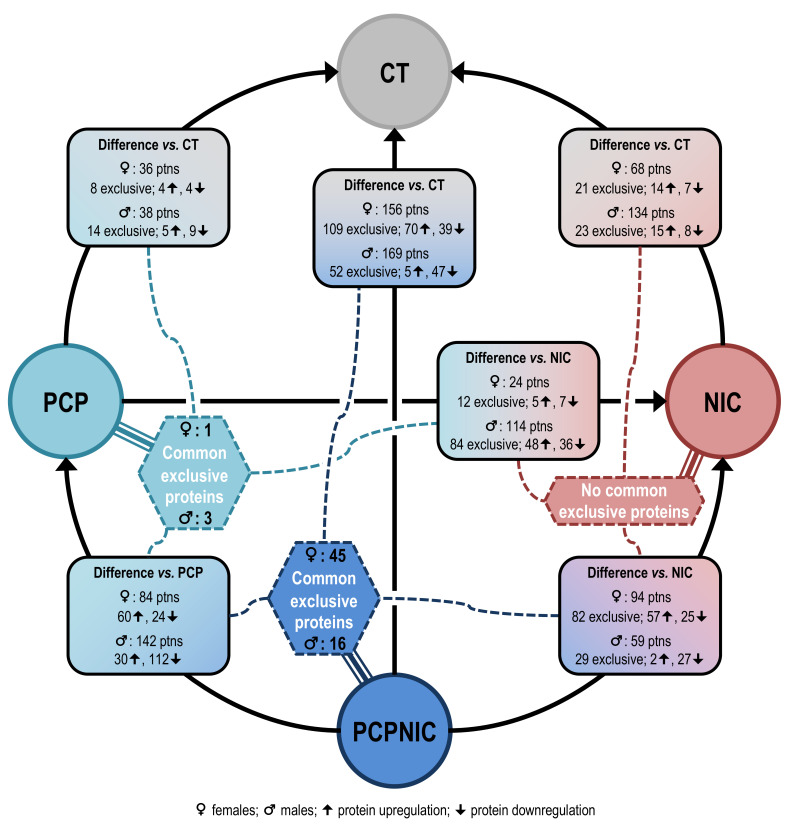
Summary of the findings for female and male mice. Circles indicate the groups used in the comparisons. Rectangles with round corners indicate the total number of deregulated proteins for each sex in the specified comparison as well as the number of deregulated proteins that were exclusive markers of a given experimental group (PCP (phencyclidine exposure), NIC (nicotine exposure) and PCPNIC) in that same comparison. The number of upregulated and downregulated marker proteins is also shown. In the hexagons, the number of proteins that were commonly deregulated in all comparisons involving the specified group is shown. Note that the group that has the highest number of commonly deregulated proteins is the PCPNIC one. Although commonly deregulated proteins were also identified regarding the PCP group, their number was considerably smaller than what was observed for the PCPNIC group. No commonly deregulated proteins were observed regarding comparisons involving the NIC group.

**Figure 2 ijms-24-14634-f002:**
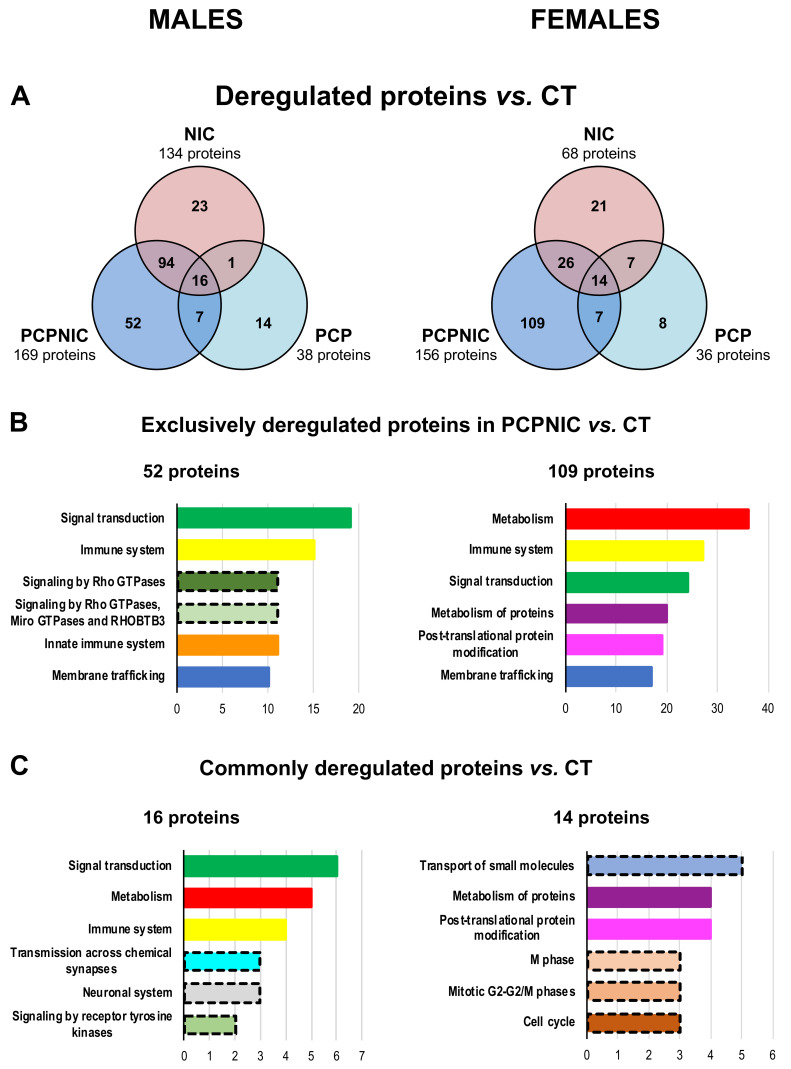
Proteomic profile of the mPFC of male (left) and female (right) mice exposed to phencyclidine (PCP), nicotine (NIC) or the combined insults (PCPNIC) when compared to controls (CT). (**A**) Venn diagrams showing deregulated proteins in NIC, PCP and PCPNIC mice when compared to CT ones. (**B**) Top 6 Reactome pathways of exclusively deregulated proteins in PCPNIC when compared to CT. (**C**) Top 6 Reactome pathways of commonly deregulated proteins in NIC, PCP and PCPNIC when compared to CT. Each color bar graph represents a specific pathway, and shades of a given color identify pathways that share a parental line. Dashed outlines represent pathways identified only once.

**Figure 3 ijms-24-14634-f003:**
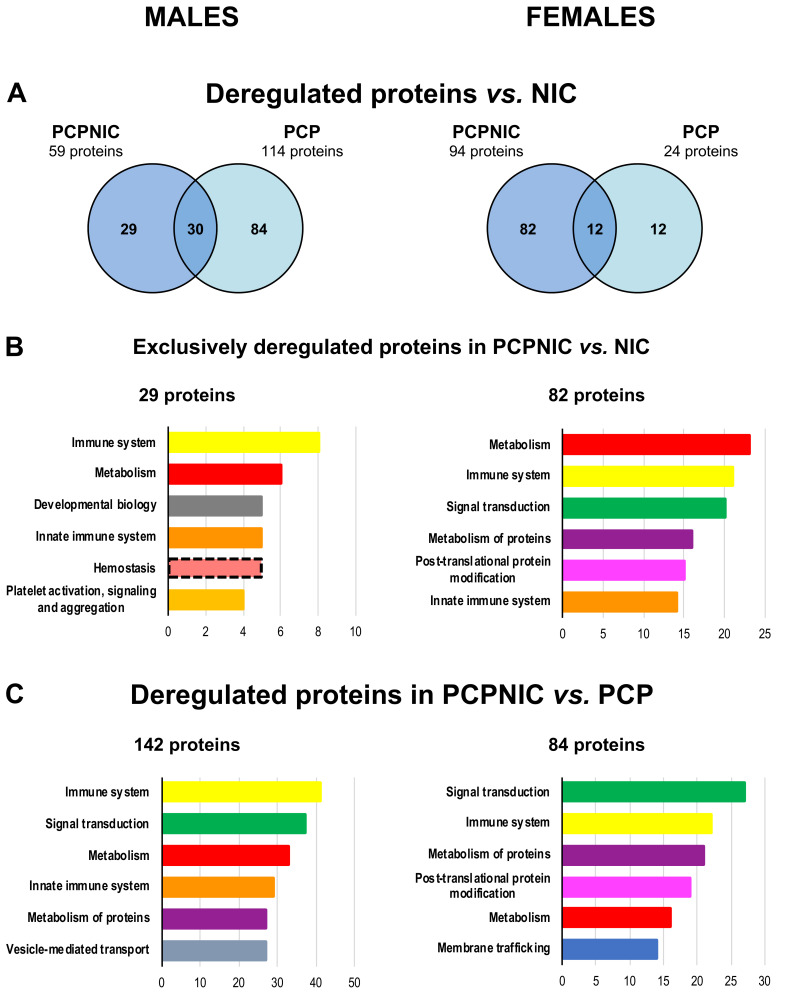
Proteomic profile of the mPFC of male (left) and female (right) mice exposed either to phencyclidine (PCP) or to the combined insults (PCPNIC) when compared to nicotine-exposed (NIC) ones (**A**,**B**) and of PCPNIC mice when compared to PCP ones (**C**). (**A**) Venn diagrams showing deregulated proteins in PCP and PCPNIC mice when compared to NIC ones. (**B**) Top 6 Reactome pathways of exclusively deregulated proteins in PCPNIC when compared to NIC. (**C**) Top 6 Reactome pathways of deregulated proteins in PCPNIC when compared to PCP. Each color bar graph represents a specific pathway, and shades of a given color identify pathways that share a parental line. Dashed outlines represent pathways identified only once.

**Figure 4 ijms-24-14634-f004:**
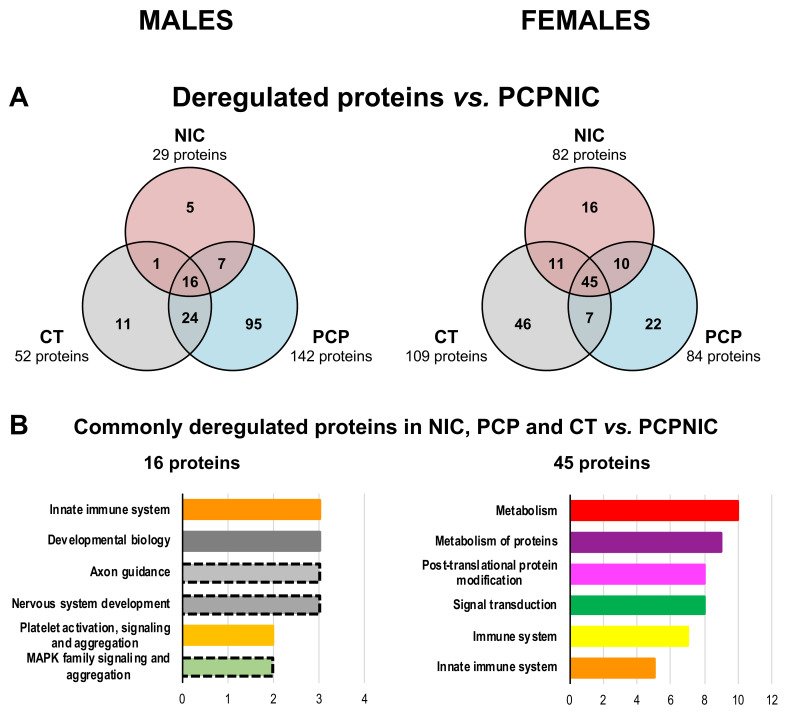
Proteomic profile of the mPFC of male (left) and female (right) mice exposed to the combined insults (PCPNIC) when compared to mice exposed to phencyclidine (PCP), mice exposed to nicotine (NIC) or the control condition (CT). (**A**) Venn diagrams showing deregulated proteins in PCPNIC mice when compared to CT, NIC and PCP ones. (**B**) Top 6 Reactome pathways of commonly deregulated proteins in PCPNIC when compared to CT, NIC and PCP ones. Each color bar graph represents a specific pathway, and shades of a given color identify pathways that share a parental line. Dashed outlines represent pathways identified only once.

**Figure 5 ijms-24-14634-f005:**
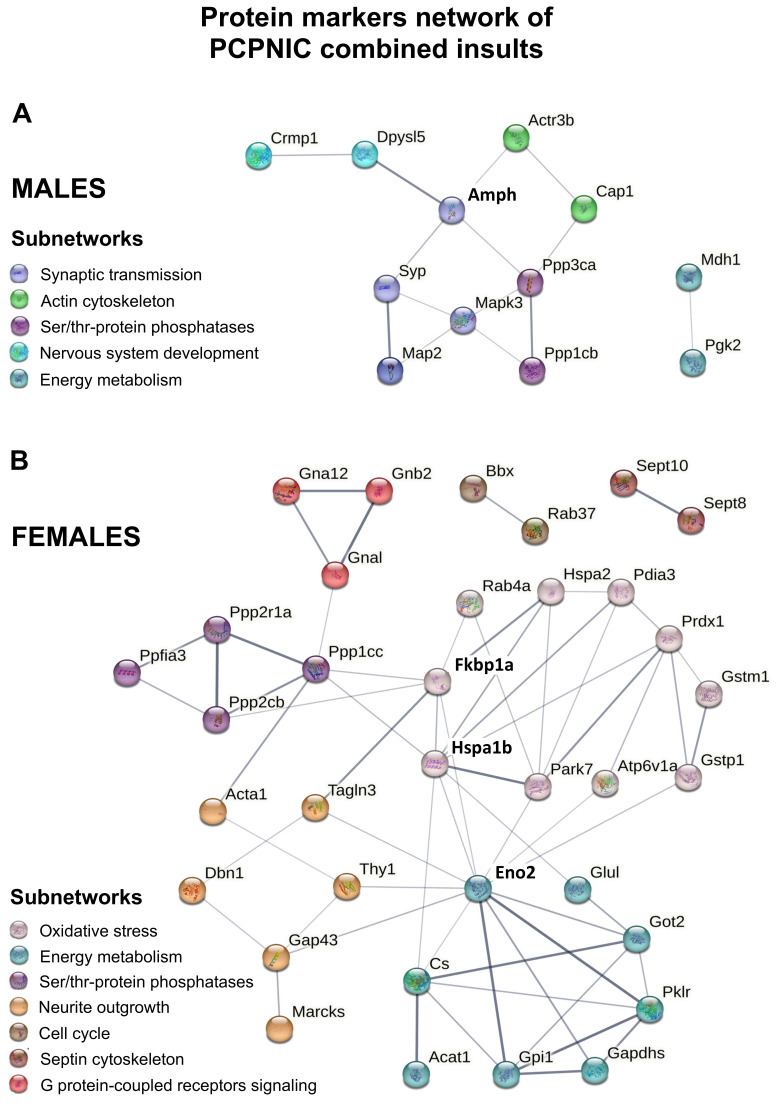
Interactome map showing protein markers of PCPNIC male (**A**) and female (**B**) mPFC. Proteins were grouped into different subnetworks so that each color represents one subnetwork. Complete lists of these proteins, organized in subnetworks, are presented in Table 1 and Table 2.

**Figure 6 ijms-24-14634-f006:**
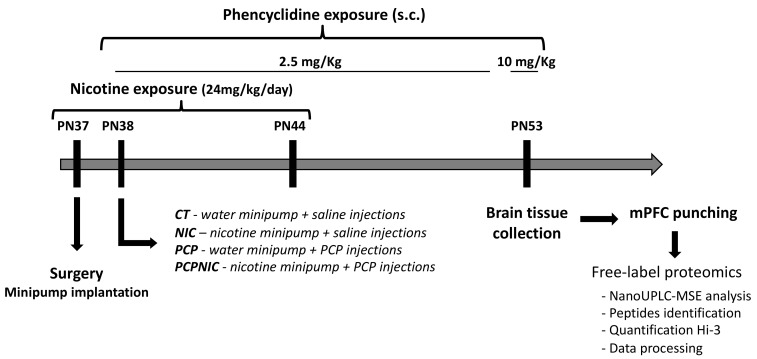
Timeline of experimental design. CT: control group; mPFC: medial prefrontal cortex; NIC: nicotine-exposed group; PCP: phencyclidine-exposed group; PCPNIC: phencyclidine- and nicotine-exposed group; PN: postnatal day.

**Table 1 ijms-24-14634-t001:** Proteins markers of PCPNIC male mice organized in interactome map subnetworks.

Subnetwork	Protein Name	Gene Name	PCPNIC vs. CT	PCPNIC vs. NIC	PCPNIC vs. PCP
**Synaptic transmission**					
	Amphiphysin	Amph	Down	Down	Down
	Synaptophysin	Syp	Down	Down	Down
	Microtubule-associated protein 2	Map2	Down	Down	Down
	Mitogen-activated protein kinase 3	Mapk3	Down	Down	Down
**Actin cytoskeleton**					
	Actin-related protein 3B	Actr3b	Down	Down	Down
	Adenylyl cyclase-associated protein 1	Cap1	Down	Down	Down
**Serine/threonine-protein phosphatases**					
	Serine/threonine-protein phosphatase 2B catalytic subunit alpha isoform	Ppp3ca	Down	Down	Down
	Serine/threonine-protein phosphatase PP1-beta catalytic subunit	Ppp1cb	Down	Down	Down
**Nervous system development**					
	Dihydropyrimidinase-related protein 5	Dpysl5	Down	Down	Down
	Dihydropyrimidinase-related protein 1	Crmp1	Down	Down	Down
**Energy metabolism**					
	Malate dehydrogenase_cytoplasmic	Mdh1	Down	Down	Down
	Phosphoglycerate kinase 2	Pgk2	Down	Down	Down

**Table 2 ijms-24-14634-t002:** Proteins markers of PCPNIC female mice organized in interactome map subnetworks.

Subnetwork	Protein Name	Gene Name	PCPNIC vs. CT	PCPNIC vs. NIC	PCPNIC vs. PCP
**Oxidative stress**					
	Peptidyl-prolyl cis-trans isomerase FKBP1A	Fkbp1a	Up	Up	Up
	Heat shock 70 kDa protein 1B	Hspa1b	Down	Down	Down
	Heat shock-related 70 kDa protein 2	Hspa2	Up	Up	Up
	Ras-related protein Rab-4A	Rab4a	Up	Up	Up
	Peroxiredoxin-1	Prdx1	Up	Up	Up
	V-type proton ATPase catalytic subunit A	Atp6v1a	Up	Up	Up
	Parkinson disease protein 7 homolog	Park7	Up	Up	Up
	Glutathione S-transferase P 1	Gstp1	Up	Up	Up
	Glutathione S-transferase Mu 1	Gstm1	Up	Up	Up
	Protein disulfide-isomerase A3	Pdia3	Up	Up	Up
**Energy metabolism**					
	Gamma-enolase	Eno2	Up	Up	Up
	Glyceraldehyde-3-phosphate dehydrogenase_testis-specific	Gapdhs	Down	Down	Down
	Glucose-6-phosphate isomerase	Gpi	Up	Up	Up
	Pyruvate kinase PKLR	Pklr	Up	Up	Up
	Aspartate aminotransferase_mitochondrial	Got2	Up	Up	Up
	Citrate synthase_mitochondrial	Cs	Down	Down	Down
	Acetyl-CoA acetyltransferase_mitochondrial	Acat1	Down	Down	Down
	Glutamine synthetase	Glul	Up	Up	Up
**Serine/threonine-protein phosphatases**					
	Serine/threonine-protein phosphatase PP1-gamma catalytic subunit	Ppp1cc	Up	Up	Up
	Serine/threonine-protein phosphatase 2A catalytic subunit beta isoform	Ppp2cb	Up	Up	Up
	Serine/threonine-protein phosphatase 2A 65 kDa regulatory subunit A alpha isoform	Ppp2r1a	Up	Up	Up
	Isoform 2 of Liprin-alpha-3	Ppfia3	Up	Up	Up
**Neurite outgrowth**					
	Thy-1 membrane glycoprotein	Thy1	Down	Down	Down
	Transgelin-3	Tagln3	Up	Up	Up
	Drebrin	Dbn1	Up	Up	Up
	Neuromodulin	Gap43	Down	Down	Down
	Myristoylated alanine-rich C-kinase substrate	Marcks	Down	Down	Down
	Actin_alpha skeletal muscle	Acta1	Up	Up	Up
**G protein-coupled receptors signaling**					
	Guanine nucleotide-binding protein subunit alpha-12	Gna12	Up	Up	Up
	Guanine nucleotide-binding protein G(I)/G(S)/G(T) subunit beta-2	Gnb2	Up	Up	Up
	Guanine nucleotide-binding protein G(olf) subunit alpha	Gnal	Up	Up	Up
**Septin cytoskeleton**					
	Septin-8	Septin8	Up	Up	Up
	Septin-10	Septin10	Down	Down	Down
**Cell cycle**					
	Ras-related protein Rab-37	Rab37	Up	Up	Up
	HMG box transcription factor BBX	Bbx	Up	Up	Up

## Data Availability

All raw data are available as Appendix A.

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
