# Peer review of "Nicotine Exposure in a Phencyclidine-Induced Mice Model of Schizophrenia: Sex-Selective Medial Prefrontal Cortex Protein Markers of the Combined Insults in Adolescent Mice"

_ijms, 2023, doi:10.3390/ijms241914634_

Round 1

Reviewer 1 Report

CORRESPONDENCE

I would not recommend the use of a YAHOO Electronic Address

ABSTRACT

Please explain the Acronym “SCHZ”

INRODUCTION

Please explain all Acronyms, such as “GABA” and “NMDA”.

The Authors should acknowledge that many patients with schizophrenia abuse nicotine in order to relief the side effects of the antipsychotics. Another issue is pleasure, as many patients suffers from anhedonia and have little more pleasures than nicotine and caffeine (not to speak about other drugs).

The Authors should acknowledge the street name for Phencyclidine such as angeldust, etc. Some background on the epidemics of it use in North America (1970’s and even more recently) would make the article more interesting for readers. A list of effects in humans, besides psychosis, and including cannibalism, could be added.  A similar approach should be done to tobacco, explaining how its abuse is one of the many causes for shorter life spam among people with schizophrenia.

MATERIAL AND METHODS

Please explain all Acronyms, such as: CEVA, UCB, VET, VENCO, MO, USA, POROS, TRAP, BEH, UERJ, CEUA, UK, jVenn (jQuery Venny), PANTHER, STRING, PRIDE, etc.

Do not forget to use symbols such as Trade Mark ™ and/or Registered ®.

FUNDING

There is no need to write BRAZIL in capital letter as it is not an acronym.

REFERENCES

Please add reference for JVENN: Philippe Bardou, Jérôme Mariette, Frédéric Escudié, Christophe Djemiel and Christophe Klopp. jvenn: an interactive Venn diagram viewer. BMC Bioinformatics 2014, 15:293 doi:10.1186/1471-2105-15-293   

Author Response

The response letter is attached.

Reviewer 2 Report

The authors describe that they used label-free proteomics to identify deregulated proteins in the medial prefrontal cortex of male and female mice modeled to schizophrenia with a history of nicotine use during adolescence. Four biomarker candidates (Amph, Fkbp1a, Hspa1 and Eno2) were identified and had been discussed in article. They had appropriateness of study design; however, the results and should be replicated and validation by other methods. My major concern is the validation.

Label-free proteomics for quantitation is popular in the proteomics community and may be the most straightforward laboratory technique in the field. However, a drawback of label-free proteomics is that run conditions (e.g. temperature, experimenter, column condition) may differ between samples. In my opinion, the major issue is that in the absence of any validated data, such as immunoblot validation and biological replication, the authors cannot conclude anything about the role of proteins in the current study in the pathophysiology of schizophrenia with a history of nicotine use during adolescence. If possible, the authors should validate proteins they found, especially Amph, Fkbp1a, Hspa1 and Eno2, by immunoblot, which confirmatory analysis will improve the level of this study in a significant manner and give a better backing for the conclusions.

Line 25: should add “(Amph)” after Amphiphysin.

The Introduction should be concise.

Author Response

 The response letter is attached.

Reviewer 3 Report

In this manuscript, authors used an adolescent mice model for of schizophrenia and nicotine misuse comorbidity. They employed label-free proteomics to identify the sex-selective protein makers and provieded the interactome map. As the authors said, this study has the potential benefit to direct future studies aiming to identify mechanism of nicotine interference in the early course of SCHZ symptoms and interventions.

The manuscript was very well designed and written, even though I found the introduction and discussion are a little long. I only have several minor comments.

1. I recommend adding the schematic diagram for study design and treatment plan.

2. Whether the data analysis was done in a double-blinded manner? I did not find the statement.

3. Please add the “A, B, C…” to the panel in Figure 2 to Figure 5, this will make clear description in Main text and Figure legends.

Author Response

The response letter is attached.

Round 2

Reviewer 2 Report

Thanks for your revision and clarification. However, the authors did not address the major point I am concerned about in the revised draft.

Major problem is label-free proteomics data validation. The label-free proteomics experiment was not validated by alternative methods such as immunoblot will lower down the data reliability a lot. Or the authors could get an independent sample set for biological replication. I might accept it if the follow up validation data or replication is solid. At least, the authors should validate 4 proteins (Amph, Fkbp1a, Hspa1 and Eno2) they found.

Figure 1, Figure 6 should be high-resolution.

Round 3

Reviewer 2 Report

Thanks for your revision and clarification. However, the authors still did not address the major point I am concerned about in the revised draft.

Actually, my major concern is label-free proteomics data validation, but the authors have not overcome this because, at present, they have not performed any of the additional experiments requested, in order to reinforce the data presented in the original submission. Thus, I do not buy their responses.
